# Pulmonary Manifestations of Babesiosis and Predictors of Mortality from a Quaternary Care Center in Westchester, New York

**DOI:** 10.3390/pathogens14040376

**Published:** 2025-04-12

**Authors:** George Williams, Luis Tatem, Kuldeep Ghosh, Arturo G. Pascual, Piotr Kapinos, Dana G. Mordue, Marc Y. El Khoury

**Affiliations:** 1Division of Infectious Diseases, Departments of Medicine, Westchester Medical Center, Valhalla, NY 10595, USA; george.williams@wmchealth.org (G.W.); luis.tatem@providence.org (L.T.); kuldeep.ghosh@wmchealth.org (K.G.); drarturogpascualjr@gmail.com (A.G.P.); piotrkapinos@gmail.com (P.K.); 2Department of Pathology, Microbiology and Immunology, New York Medical College, Valhalla, NY 10595, USA; dana_mordue@nymc.edu; 3Division of Infectious Diseases, Department of Medicine, New York Medical College, Valhalla, NY 10595, USA

**Keywords:** babesiosis, tick-borne disease, pulmonary manifestations, mortality, severe disease, predictors, parasitemia

## Abstract

Babesiosis is an emerging tick-borne illness, and pulmonary manifestations have not been well described previously. This is a single-center, retrospective study of hospitalized adults with confirmed babesiosis at Westchester Medical Center (Valhalla, NY, USA) from 2006 to 2023. Descriptive analysis was used to summarize the demographic and clinical data. Logistic regression was used to determine factors associated with severe disease and mortality. A total of 147 cases were reviewed; 63 were severe, 41 were admitted to the intensive care unit, and ten died. Respiratory symptoms were observed in 42% of cases (cough, dyspnea, oxygen supplementation, respiratory failure). A total of 55 patients had an abnormal chest radiograph (infiltrates, pleural effusions), and 75% of severe cases had parasitemia ≥ 10%. Factors associated with severe cases included age ≥ 70 years, severe dyspnea, radiographic abnormalities, white blood cell count > 5000 cells/dL, hemoglobin 8 g/dL or less, and increased creatinine. Lyme seropositivity was common (28%). Exchange transfusion did not correlate with outcome. Increased mortality was more likely in patients 70 years or older or in those having any of the following: encephalopathy, acute kidney injury, severe dyspnea on admission, abnormal chest radiograph, and requiring intubation. In conclusion, pulmonary symptoms including abnormal chest radiograph are not uncommon in babesiosis and are associated with worse outcomes.

## 1. Introduction

Babesiosis in humans is a tick-borne malarial-like illness caused by four species of the intra-erythrocytic protozoa of the genus *Babesia* (*B. microti*, *B. duncani*, *B. divergens*, *B. venatorum*). In the United States, primarily between May and October, *B. microti* is responsible for the majority of human cases, while in Europe most cases are due to *B. divergens* [1,2]. *B. microti* zoonotic reservoirs are the white-footed mouse and the meadow vole [3]. The parasite is primarily transmitted by the bite of *Ixodes scapularis* (blacklegged tick) [4]. Cases have also been reported secondary to perinatal transmission, blood transfusions, and solid organ transplantation [5]. The severity of the disease can range from asymptomatic illness to life-threatening infection [1]. The blacklegged tick’s geographical range covers the Northeast United States, making babesiosis endemic to many states, five of which are located in the Northeast (MA, CT, RI, NY, NJ) and two in the upper Midwest (MN, WI). More cases are now also reported all along the northeastern seaboard and inland, ranging from Maine to Maryland [2]. During 2011–2019, a total of 16,456 cases of babesiosis were reported to the Centers for Disease Control and Prevention (CDC) by 37 states, with New York state averaging the most cases per year over this time period [6]. Furthermore, between 2011 and 2019 babesiosis cases increased significantly in most states (except WI, MN), including a 58.3% increase in New York [6]. Groups at risk for severe infection include the elderly, immunosuppressed patients, and patients that are asplenic [1]. Typical symptoms of babesiosis include fever, arthralgias, fatigue, and headaches [1,7]. However, due to its non-specific presentation, babesiosis is often misdiagnosed [8]. Recognizing the clinical features of severe disease and early diagnosis could prevent complications, especially in vulnerable groups. Furthermore, the pulmonary manifestations of this infection were not well highlighted previously. In this study, we aim to describe the clinical manifestations of babesiosis, including its pulmonary involvement and radiographic findings and study the different factors associated with severe disease and increased mortality.

## 2. Materials and Methods

We conducted a retrospective chart review of all patients diagnosed with babesiosis admitted to Westchester Medical Center between July 2006 and August 2023. The study was conducted according to the guidelines of the Declaration of Helsinki and approved by the Institutional Review Board (IRB) of New York Medical College, protocol code L,11-845 on 10 March 2017. An informed consent waiver was approved by the IRB since the study is a chart review, involves minimal risk for the subjects, and personal identifiers remain confidential and not shared with the results of the study.

Patients were diagnosed with babesiosis and included in the study if they presented with systemic symptoms and were found to have either a positive blood parasite smear for intra-erythrocytic parasite consistent with babesia species (in the absence of recent travel to endemic area for malaria) or positive *B.microti* by molecular testing. Patients were excluded if they were less than 18 years old, had a hospital stay of less than 24 hours or only visited the emergency room, or babesiosis was transfusion-related. Recurrent cases were counted only once. We collected demographic data including age, gender, county of residence, date of admission, occupational risk, as well as comorbidities (presence of active malignancy, immunosuppression). In this study, immunosuppression was defined as follows: taking immunosuppressive medication such as prednisone of 15 mg or more or other immunomodulatory agent, having end-stage organ failure, uncontrolled diabetes mellitus, asplenia, or having an active malignancy. We also recorded clinical characteristics of patients, including intensive care unit (ICU) admission; length of stay; splenic infarcts or rupture; central nervous system symptoms, such as headaches, dizziness, blurry vision and altered mentation (confusion, delirium); respiratory symptoms, such as cough, dyspnea and severe dyspnea; coinfection; pulmonary radiographic findings; need for exchange transfusion; acute kidney injury (increased serum creatinine by 1.5 times compared to baseline); and death or discharge alive from the hospital. Lyme coinfection was considered if patients had a rash consistent with erythema migrans or if *B.burgdorferi* antibodies were positive on enzyme immunoassay and the presence of at least five out of ten IgG and or two out of three IgM bands on western blot. Coinfection with anaplasmosis was defined as a positive PCR for *Anaplasma phagocytophilum*. 

Additionally, we collected laboratory values such as highest parasitemia, lowest hemoglobin, and liver function tests. Lowest hemoglobin was defined as the lowest hemoglobin level during the hospital stay for babesiosis. Serum levels of creatinine, haptoglobin, lactate dehydrogenase, white blood cells count, percentage of neutrophils and lymphocytes, and platelets count were also recorded. We recorded the type and duration of initial and final antimicrobial given as well. In this study, we defined severe babesiosis as the presence or requirement of one of the following: ICU admission, acute kidney injury requiring hemodialysis, parasitemia over 10%, acute change in mental status, exchange transfusion, severe coagulopathy, or respiratory failure requiring intubation. 

We used IBM SPSS 27 (IBM, Armonk, NY, USA) for all statistical analyses. For descriptive analysis we used means and standard deviations to analyze nominal data, and we performed chi-square tests, and *t*-tests for continuous parametric data. To look at factors associated with severe infection and increased mortality, we calculated odds ratios and 95% confidence intervals, using binomial logistic regressions. If odds ratios were not obtainable due to perfect separation, we reported the relative risk and 95% confidence intervals by using a chi-square test. All comparisons were two-tailed and we considered *p*-values < 0.05 significant.

## 3. Results

### 3.1. Demographics and Epidemiology

The mean age was 64.1 ± 16.2 years; 69% were males. On average, patients with severe babesiosis were 12 years older than non-severe cases (Table 1). The majority of patients resided in Westchester, Dutchess, Orange, or Putnam County (Figure 1A,B). Most (77%) patients were admitted in July and August (Figure 1C). A heatmap is also provided below (Figure 2) for comparison to show the distribution of cases across 40 states. Only 18% of patients could recall being bitten by a tick within the 2 weeks prior to admission (Table 1).

### 3.2. Clinical Findings

Clinical manifestations, complications, treatments, and length of stay are shown in Table 1 Of the 147 cases, 63 (43%) were considered severe. Forty-one patients (28%) required admission to the intensive care unit. Other than myalgia, fatigue, and arthralgia, common clinical features included fever (72%); central nervous system manifestations (35%); such as altered mental status, headaches, confusion, and respiratory symptoms, (42%) such as cough and/or dyspnea or severe dyspnea requiring oxygen supplementation; and respiratory failure requiring mechanical ventilation. The presence of severe dyspnea requiring oxygen supplementation was more likely to be seen in severe than non-severe cases. There were 23 patients (16%) deemed to be immunosuppressed, and 11 of these patients had asplenia. Coinfection with Lyme disease was present in 28% of patients. Eleven (7.5%) patients required hemodialysis due to severe acute kidney injury. Five (3.4%) patients had splenic rupture on admission; all required ICU monitoring.

### 3.3. Laboratory and Radiographic Findings

Laboratory and radiographic results on admission are shown in Table 2.

Patient results are divided into three different columns showing the results for the whole cohort in one and the rest divided between severe and non-severe cases. The mean white blood cell count was 7.7 × 10^3^ cells/mm^3^, and hemoglobin was 8.7 g/dL with a platelet count of 85.6 × 10^3^ cells/mm^3^. Liver function tests were on average two–three times above the normal limits. Laboratory values that were significantly different between the severe and non-severe cases include a lower hemoglobin and percentage of lymphocytes, a higher white blood cell count, percentage of neutrophils, a higher creatinine, liver enzymes including total bilirubin, and lactate dehydrogenase. Most patients (76%) had a parasitemia level below 10%, and 56 (38%) patients had a parasitemia below 1%. Only 49% of patients with severe babesiosis had a parasitemia level more than 10%. There were 55 patients (37%) with chest radiograph abnormalities such as lung infiltrates with or without effusions, which was significantly more common in severe babesiosis. An abnormal chest radiograph was seen in 37% of cases. The most common finding in chest radiographs was the presence of infiltrates in 22%, which was significantly different between both groups studied.

### 3.4. Treatment and Outcome

Sixty-three (43%) patients had severe babesiosis, of whom 41 (65%) were admitted to the ICU. Eleven (8%) patients needed mechanical ventilation, and 10 (6.8%) patients died. The majority of patients (75%) received azithromycin and atovaquone (AA) as their final course of treatment, and 8% received clindamycin and quinine (CQ). Severe cases were more likely to have received clindamycin and quinine (*p* = 0.02) and to have a longer length of stay of 9.5 days compared to 6.3 days on average (*p* < 0.001). In a subgroup analysis of all patients with severe babesiosis, by comparing patients who died versus patients who survived, we found no statistically significant difference in the initial or final antimicrobial regimen used, whether it was AA or CQ among the two groups. A logistic regression analysis was also performed and found that patients with severe babesiosis who were treated with CQ were not more likely to survive compared to those who were treated with AA with the following odds ratios (OR 1.2 [95% CI, 0.25–5.70] for initial regimen) and (OR 2.8, [95% CI 0.43–18.68] for final regimen). Thirty-one (21%) patients needed a red blood cell exchange transfusion (ET). In the subgroup of patients with severe disease, we found no difference in mortality in patients who received ET compared to those who did not (*p* = 0.31).

A list of factors that are predictive of severe babesiosis are shown in Table 3 along with their odds ratios.

Clinical conditions associated with severe babesiosis included older age with an odds ratio of 9.7 for patients more than 70 years old, higher parasitemia levels, acute kidney injury (OR 4.62 [95% CI, 2.23–9.56]), and splenic rupture (OR 1.11 [95% CI, 1.01–1.22]). The mean age of the five patients who had splenic rupture was 76 years (range of 64–86 years); none had immunosuppression or major comorbidities. Four of the five had a parasitemia level of 1% or less, and one had 8%. The two oldest patients with splenic rupture were 86 years old; both developed respiratory failure requiring mechanical ventilation, and one of them also developed AKI and died. The remaining three had transient hypotension and tachycardia requiring ICU monitoring.

Patients with severe infection were more likely to complain of severe dyspnea or to have pulmonary infiltrates on chest radiographs. Severe cases had a mean hemoglobin of 8 g/dL, lactate dehydrogenase (LDH) of 1191 (U/L), total bilirubin of 4.8 mg/dL, a mean white blood cell count of more than 5 × 10^9^ cell/L, and a mean creatinine of 1.8 mg/dL. A total of ten patients died (7%), nine (14%) of whom had severe babesiosis (OR 13.83 [95% CI, 1.7–112.31]). Factors that predicted mortality are shown in Table 4.

Age was a significant predictor of mortality. Patients aged 70 years old or more were six times more likely to die (OR 5.96 [95% CI, 1.22–29.15]). Acute change in mental status, while being a criterion for a severe case, was also a predictor of mortality (OR 12.20 [95% CI, 2.93–50.83]). Acute kidney injury and requiring hemodialysis were also significant predictors with an odds ratio of 21 [95% CI, 2.5–165]. Having severe dyspnea and abnormal chest imaging, or signs of acute heart failure and having received antibiotics for pneumonia were more frequently associated with increased mortality.

## 4. Discussion

This retrospective chart review study is the largest single-center study to describe the clinical manifestations and outcome of babesiosis in the lower Hudson Valley of New York State where babesiosis is endemic, along with Lyme disease and anaplasmosis. All three diseases are transmitted through the bite of the deer tick I. *scapularis* [3]. Patients commonly present with a single infection, but coinfections should be also considered in certain cases where presentation is atypical or more severe [11]. Babesiosis was one of the emerging tick-borne infections in this region during 2011–2019, as the incidence of babesiosis in the U.S. significantly increased in northeastern states. Some states that were not considered endemic still exhibited a significant increase in reported cases similar to or even higher than those classified as endemic states [6]. Babesiosis can result in an asymptomatic infection or minimal symptoms in 40% of children and 20% of young and healthy adults, for which patients seldom seek medical care [12]. In the elderly and patients with comorbidities, the infection may result in significant morbidities and even death, with a rate ranging from 1.6% to 21% [7,13,14,15].

Our study highlights several important points that will assist clinicians with their inclusion of babesiosis in the differential diagnosis of a febrile illness in an endemic area. Similar to previous studies, we noted that in our cohort the average age of hospitalized patients with babesiosis was 64 years old, with more severe presentation in patients 70 years or older [5,7,14]. In one study, however, the average age was 53 years, with severe presentation reported in patients older than 60 years old [16]. Males were two times more common than females, which is explained by the fact that more men than women are engaged in outdoor activities leading to frequent tick exposures. Only 18% of patients recalled having had a tick bite, which is similar to what was reported in children in one study [17] and within a range of 9% to 40% reported in adults in prior studies [5,7,13,18]. Thus, the absence of a history of a tick bite should not be used as a factor to determine the likelihood of tick-borne illness. Patients presented with a wide range of clinical symptoms, with a high fever (mean of 38.9 degrees Celsius) being the most common manifestation in all patients regardless of severity. Other symptoms have been described in detail elsewhere [7].

In this study, we focused primarily on the respiratory symptoms associated with babesiosis, which were cough in 23% (similar to what was reported by White et al. and Hatcher et al.) [7,16]), dyspnea in 22%, and a combination of both in 4%. Severe dyspnea with hypoxemia requiring oxygen supplementation and respiratory failure requiring mechanical ventilation were reported in 7.5% of the cases, respectively. This finding was similar to the 6.3–8% rate of acute respiratory distress syndrome and slightly higher than 3.1% to 5.8% intubation rate reported by White et al. and Mareedu et al. [7,18] and close to the 6.8% rate that was recently described in the largest epidemiological study of hospitalized patients with babesiosis [14]. Acute congestive heart failure was seen in 7.5% in our study compared to 3.5 to 10.9% elsewhere [7,14]. Neurologic manifestations were seen in 35% of cases and ranged from headaches and confusion to altered mental status, which were more common in patients with severe disease. It is worth noting that a change in mental status specifically was seen in 70% of patients who expired compared to 16% in those who survived. Acute kidney injury was observed in 35% of cases, with acute renal failure requiring hemodialysis in 7.5%. The rate of this complication had been variable in the literature, with a range of 4.3 to 20.4% of reported renal failure and 1.4% to 8.6% of cases requiring hemodialysis [7,14,15,16,18]. The differences in these reported rates of complications is partly related to the definition used in each study. The mean length of stay ranged from 6 days in non-severe cases to 10 days in severe cases compared to a range of 7 to 14 days in other studies with few outliers beyond 15 days [5,7,14,16]. 

From the laboratory perspective, compared to non-severe cases, severe cases had lower hemoglobin levels, platelet counts, and worse transaminitis, hyperbilirubinemia, and LDH levels and a higher white blood cell count. None of the non-severe cases had a parasitemia of more than 10%. These clinical findings have not been clearly reported in prior studies [5,7]. The mean parasitemia level in the severe cases was 12.6% with a standard deviation of 12.5 indicating significant variability compared to 1.8% with a 2.2 standard deviation in the non-severe group. These rates are similar to what was previously reported, with a range of 8 to 10.1% reported in severe cases. In one study, the median parasitemia was 2% in all cases, with a range of 1 to 80%, with severe cases having levels ranging between 2% and 50% [13]. In two other studies, there was no difference found between severe and non-severe cases which had a parasitemia level more than 10% [16,18]. These observations suggest that severe disease is not necessarily always associated with a parasite level of more than 10% due to other contributing factors related to host immune status and associated comorbidities. The coinfection rate with Lyme disease noted in our cohort was 28% similar to what was reported previously [5,18]. This elevated rate of coinfection may be partly due to false-positive serological tests in some cases, as previously reported [19], since many patients did not have other manifestations consistent with Lyme disease, such as erythema migrans, carditis, or cranial nerve palsies. Some patients were empirically treated for Lyme disease pending serological tests. A red blood cell exchange transfusion (ET) was performed in 31 (21%) cases which met the criteria for severe disease due to either organ failure or parasitemia of more than 10%. The reported rate of ET ranged from 4.3% to 20.6% [5,7,16]. This increased rate in our study was not unexpected given the older average age compared to other studies, with more than 30% of patients being older than 70 years. In the subgroup of patients with severe disease, we found no difference in mortality between those who had ET and those who did not. This finding was similar to a recent retrospective study of 19 patients where neither the post-ET parasitemia nor the change in parasitemia correlated with the length of stay or mortality rate, suggesting that repeat ETs should only be based on clinical status of the patient and not necessarily on a fixed parasitemia level [20]. 

In our study, we noted that severe cases were more likely to have received clindamycin and quinine sulfate (CQ) initially compared to azithromycin and atovaquone (AA) and the opposite for non-severe cases. This difference in type of therapy based on severity of the disease seen in our study was driven by the belief of most practitioners that the combination of CQ in severe cases was superior to the combination of AA. The practice and standard of care had changed after the publication of the updated guidelines on the treatment of babesiosis in November 2020 [21]. The updated guidelines recommended the use of a higher dose of azithromycin in addition to atovaquone. This regimen is used even in severe babesiosis as a first-line therapy due to its better tolerability and good response rate. CQ is currently reserved for cases which fail to respond adequately to the first line of therapy [21]. It is also worth noting that the type of therapy used initially in our study had changed during the course of hospitalization in some patients due to either side effects from CQ, lack of adequate response to AA, or improvement where patients who were on CQ were changed back to AA for better tolerability. Other combinations also were used due to either a history of an allergy to one of the drugs, new onset side effects, or lack of response to standard therapy. In our study, similarly to what was reported elsewhere [22], we also found out that using AA in severe cases was not more likely to be associated with increased mortality compared to using CQ, even though in some cases the change may be warranted based on a case-by-case decision.

In our study, immunosuppression and splenectomy were not found to be associated with severe disease or increased mortality, which was similar to the findings on the multivariate analysis reported by White et al. and Meldrum et al. [7,13]. This finding could be due to a sample bias or to the fact that immunosuppression and/or splenectomy plays a more important role in prolonged clearance of the infection and frequent relapse rather than the severity of infection. Infection severity and mortality due to babesiosis may be more linked to the cytokine storm triggered by the infection, leading to end organ damage. One study has reported a significant association between splenectomy or having an autoimmune disease and severity of the disease, without any effect on mortality [18]. Another case-control study comprised mainly of patients receiving or having received rituximab for B cell lymphoma in the case group has reported a significant association with severe disease and mortality; however, this study had only 14 patients in that arm [15].

In our study, we found that patients with splenic rupture were more likely to be considered to have severe disease because the oldest two (86 years old) had subsequent respiratory failure, while the other three required a short ICU stay for monitoring. The complicated course that occurred in the oldest patients despite a low parasitemia level in this group was likely due to frailty or an unknown underlying comorbidity. Complications secondary to splenic rupture in the upper extreme of ages has been also reported by Dumic et al. [23]. In their systematic review, the median age was 56 years; the youngest was 23 years, and the oldest 85 years. Levels of parasitemia were documented in 26 of 34 cases (76%) and ranged from 0.1% to 30% (median 1%). Most patients were managed with spleen preservation [23]. In general, it is believed that around 1% of patients with babesiosis may develop splenic rupture [24]. Based on current evidence, we can say that most cases reported to date occurred in younger and healthier patients with a level of parasitemia in most cases <10% [23,24]. The mechanism of splenic rupture is thought to be related to a more robust erythrophagocytosis by splenic histiocytes, leading to pathological mechanical strain and rupture [1,24]. The management of atraumatic, spontaneous splenic rupture should emphasize the preservation of the spleen whenever possible in this population.

Finally, in our study the mortality rate was 7%, which is similar to the 5–6.5% range reported in two similar large studies [7,13] but significantly higher than the 1.6% reported in the recent analysis of the National Inpatient Sample database by Bloch et al. [14]. We believe the mortality rate seen at our institution is due mainly to a selection bias (sicker patients) or possibly to a delay in appropriate diagnosis and/or treatment given that many patients treated at our tertiary-care hospital were transfers from several surrounding community hospitals for a higher level of care. We do not think age was a major contributing factor in this mortality rate discrepancy because the mean age (64 years) was similar; 50% of patients were 67 years or older in the study reported by Bloch et al. [14] versus 43% of patients in our study were 70 years or older. Two other studies reported extreme mortality rates: one by Mareedu et al., with no fatalities despite having had a significant number of patients older than 70 years [18], while Krause et al. reported a fatality rate of 21% in their highly immunosuppressed arm of the study but with an overall mortality of 5% [15]. This significant variability in mortality rate in these two studies seems to be related to several confounding factors depending on the population studied.

Our study sheds light on multiple aspects of babesiosis that have not been explored previously. It helps provide guidance to a broad range of providers, whether in the emergency room, in the outpatient department, or infectious diseases specialists. The limitations of the data presented include the fact that this was a retrospective chart review study, with selection bias for more severe presentations and older populations and that the data came from a single site. This will only provide insights into this specific population and not include the whole spectrum of disease presentation. 

## 5. Conclusions

The diagnosis of babesiosis can be easily missed or often delayed given the non-specific manifestations upon presentation. The presence of pulmonary symptoms and abnormal chest radiographs are not uncommon and are often misleading. A summer respiratory illness with abnormal liver function or thrombocytopenia should always prompt testing for babesiosis. The presence of any organ dysfunction, an abnormal chest imaging, or severe dyspnea regardless of the parasitemia level was found to be associated with a worse outcome and increased mortality that warrants hospitalization and close monitoring. Lyme coinfection maybe over-diagnosed due to possible false-positive serological testing and warrants further study. In severe disease, AA is well-tolerated and not likely to be associated with increased mortality. The role of exchange transfusion in severe disease with a parasitemia level of 10% or less is unclear and warrants further study.

## Figures and Tables

**Figure 1 pathogens-14-00376-f001:**
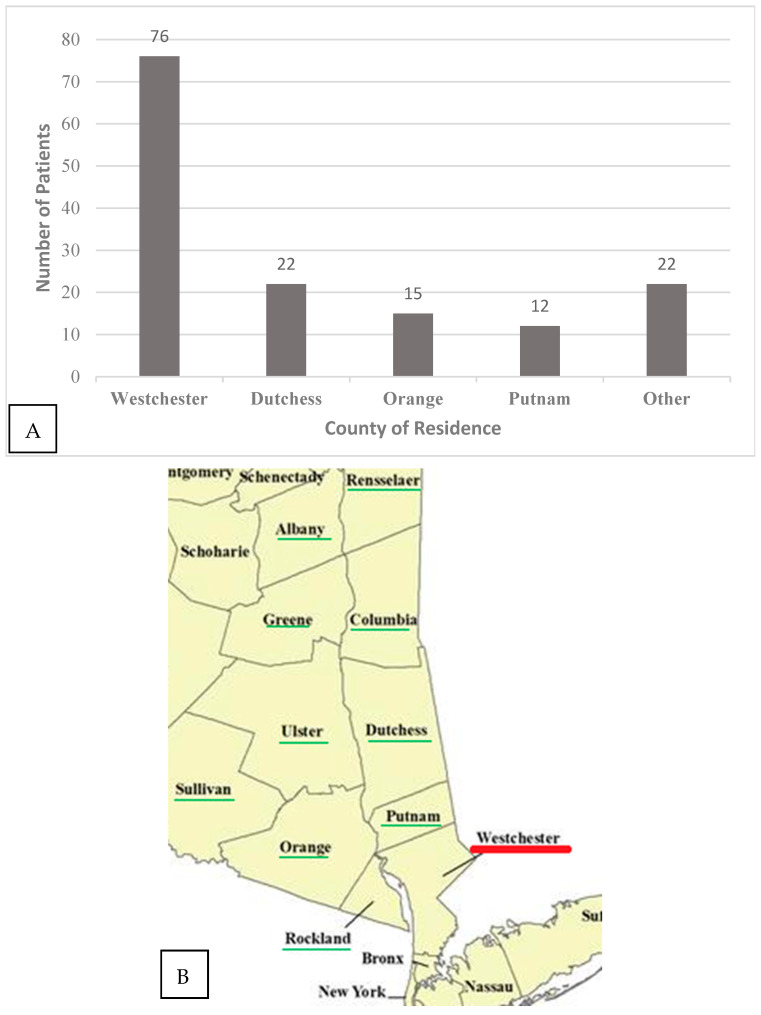
(**A**) County of residence for 147 babesiosis cases in New York State, admitted to Westchester Medical Center (WMC). (**B**) Map of counties in the Hudson Valley of New York State [9]. Underlined are the counties that are part of the Hudson Valley. The red line indicates the county where WMC is located. (**C**) Month of admission for 147 babesiosis cases in New York State admitted to WMC.

**Figure 2 pathogens-14-00376-f002:**
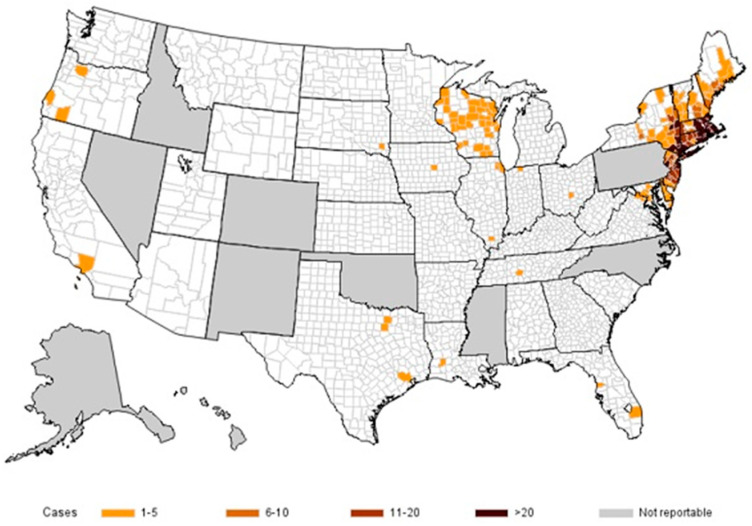
Heatmap of the United States showing the number of reported cases of babesiosis by county of residence for 40 states in 2020. The map is developed by the CDC’s National Center for Emerging and Zoonotic Infectious Diseases (NCEZID) and is available on the agency website for no charge [10].

**Table 1 pathogens-14-00376-t001:** Demographic (1a) and clinical (1b) characteristics of 147 babesiosis cases in New York State. ad-mitted to WMC.

	Patients, No. (%)			
(1a) Demographic Characteristics	All (n = 147)	Non-Severe (n = 84)	Severe (n = 63)	*p*
Age, mean ± SD, years	64.1 ± 16.2	58.8 ± 16.9	71.2 ± 12.1	<0.001
Sex, no. (%)				0.54
Male	102 (69)	60 (71)	42 (67)	
Female	45 (31)	24 (29)	21 (33)	
History of tick bite	27 (18)	15 (18)	12 (19)	0.85
**(1b) Clinical Characteristics**	**All (n = 147)**	**Non-Severe (n = 84)**	**Severe (n = 63)**	** *p* **
ICU admission	41 (28)	0 (0)	41 (65)	<0.001
Central nervous system involvement	51 (35)	15 (18)	36 (57)	<0.001
Fever (°C), mean ± SD	38.9 ± 0.6	38.9 ±0.6	39 ± 0.6	0.40
Presence of fever	106 (72)	66 (79)	40 (63)	0.044
All respiratory symptoms	62 (42)	32 (38)	30 (48)	0.25
Cough	23 (16)	17 (20)	6 (10)	0.08
Dyspnea	22 (15)	9 (11)	13 (21)	0.10
Cough and dyspnea	6 (4)	4 (5)	2 (3)	0.63
Severe dyspnea	11 (7.5)	2 (2)	9 (14)	0.007
Intubation	11 (7.5)	0 (0)	11 (17)	<0.001
Acute kidney injury	51 (35)	17 (20)	34 (54)	<0.001
Required hemodialysis	11 (7.5)	0 (0)	11 (17)	<0.001
Congestive heart failure	11 (7.5)	5 (6)	6 (10)	0.352
Immunosuppression	23 (16)	12 (14)	11 (17)	0.60
Splenectomy	11 (7.5)	5 (6)	6 (10)	0.35
History of malignancy	21 (14)	9 (11)	12 (19)	0.16
Lyme coinfection	41 (28)	22 (26)	19 (30)	0.32
Exchange transfusion	31 (21)	0 (0)	31 (49)	<0.001
Azithromycin + atovaquone	110 (75)	69 (82)	41 (65)	0.02
Clindamycin + quinine	12 (8)	3 (4)	9 (14)	0.02
Other combinations	25 (17)	12 (14)	13 (21)	0.311
Length of stay, days, mean ± SD	7.7 ± 5.8	6.3 ± 5.3	9.5 ± 5.8	<0.001
Length of treatment, days, mean ± SD	15.2 ± 13.3	14.4 ± 10.0	16.3 ± 17.1	0.41

SD = standard deviation, No = number of patients, ICU = intensive care unit, °C = degree Celsius.

**Table 2 pathogens-14-00376-t002:** Laboratory and radiographic findings for 147 babesiosis cases in New York State admitted to WMC.

	Patients, Mean ± SD			
(2a) Laboratory Values	All (n = 147)	Non-Severe (n = 84)	Severe (n = 63)	*p*
Lowest hemoglobin (g/dL)	8.7 ± 2.4	9.3 ± 2.7	8.0 ± 1.6	<0.001
Platelets (×10^9^ cells/L)	85.6 ± 72.1	90.9 ± 79.4	78.3 ± 60.9	0.30
WBC (×10^9^ cells/L)	7.7 ± 7.6	6.3 ± 3.2	9.6 ± 10.8	0.008
Creatinine (mg/dL)	1.4 ± 1.2	1.0 ± 0.4	1.8 ± 1.6	<0.001
AST (U/L)	101.7 ± 88.6	80.6 ± 67.6	130.4 ± 104.7	<0.001
ALT (U/L)	84.1 ± 72.8	74.7 ± 76.2	96.9 ± 66.4	0.07
Alkaline phosphatase (U/L)	111.9 ± 57.5	114.1 ± 56.0	108.9 ± 59.8	0.59
LDH (U/L)	971.7 ± 612.4	772.6 ± 527.9	1191.0 ± 628.0	<0.001
Total bilirubin (mg/dL)	3.1 ± 4.5	2.2 ± 1.9	4.5 ± 6.4	0.002
Neutrophils (%)	62.9 ± 15.9	57.9 ± 13.9	69.8 ± 15.9	<0.001
Lymphocytes (%)	22.8 ± 15.1	26.0 ± 13.8	18.4 ± 15.8	0.003
Parasitemia (%)	6.6 ± 10.1	1.8 ± 2.2	12.6 ± 12.5	<0.001
**Parasitemia Groupings**	**Patients, No. (%)**			
Less than 1	56 (38)	40 (48)	16 (25)	0.01
1 up to 4	32 (22)	27 (32)	5 (8)	<0.001
From 4 to 10	24 (16)	13 (15)	11 (17)	0.75
Greater than 10	31 (21)	0 (0)	31 (49)	<0.001
Unknown	4 (3)	4 (5)	0 (0)	
**(2b) Radiographic Findings**	**Patients, No. (%)**			
Chest X-ray abnormalities	55 (37)	16 (19)	39 (62)	<0.001
Lung infiltrates	32 (22)	9 (11)	23 (37)	<0.001
Pleural effusion	10 (7)	5 (6)	5 (8)	0.64
Lung infiltrates and pleural effusions	6 (4)	1 (1)	5 (8)	0.04

**Table 3 pathogens-14-00376-t003:** Logistic regression analysis for factors associated with severe disease.

	Patients, No. (%)			
Clinical and Laboratory Factors	Non-Severe (n = 84)	Severe (n = 63)	*p*	Odds Ratio (95% CI)
Age ≥ 70 years	25 (30)	38 (60)	<0.001	9.73 (3.34–28.34)
Splenic rupture	0 (0)	5 (8)	0.004	1.11 (1.01–1.22)
Acute kidney injury	17 (20)	34 (54)	<0.001	4.62 (2.23–9.56)
Chest X-ray abnormalities	16 (19)	39 (62)	<0.001	7.42 (3.49–15.80)
Lung infiltrates	9 (11)	23 (37)	<0.001	7.11 (2.874–17.596)
Lung infiltrates and pleural effusion	1 (1)	5 (8)	0.019	13.91 (1.54–125.47)
Severe dyspnea	2 (2)	9 (14)	0.016	6.83 (1.42–32.85)
Mortality	1 (1)	9 (14)	0.014	13.83 (1.70–112.31)
	**Patients, Means ± SD**			
**Clinical and Laboratory Factors**	**Non-Severe (n = 84)**	**Severe (n = 63)**	** *p* **	**Odds Ratio (95% CI)**
Age, years	58.8 ± 16.6	71.2 ± 12.1	<0.001	1.06 (1.03–1.09)
Highest parasitemia	1.8% ± 2.2%	12.6% ± 12.5%	<0.001	1.33 (1.19–1.49)
Lowest hemoglobin (g/dL)	9.3 ± 2.7	8.0 ± 1.6	0.001	0.74 (0.62–0.89)
Lactate dehydrogenase (U/L)	772.6 ± 527.9	1191.0 ± 628.0	<0.001	1.001 (1.001–1.002)
Total bilirubin (mg/dL)	2.2 ± 1.9	4.8 ± 6.4	0.013	1.23 (1.05–1.46)
Aspartate aminotransferase (U/L)	80.6 ± 67.6	130.4 ± 104.7	0.002	1.007 (1.003–1.012)
White blood cells (×10^9^ cells/L)	6.3 ± 3.2	9.6 ± 10.8	0.008	1.13 (1.03–1.24)
White blood cells ≥ 5.0 × 10^9^ cells/L	50 (60)	48 (76)	0.036	2.18 (1.05–4.49)
Neutrophils (%)	57.9 ± 13.9	69.8 ± 15.9	<0.001	1.06 (1.03–1.10)
Lymphocytes (%)	26.0 ± 13.8	18.4 ± 15.8	0.005	0.96 (0.93–0.99)
Creatinine (mg/dL)	1.0 ± 0.4	1.8 ± 1.6	<0.001	3.13 (1.63–5.98)

No. = number of patients, CI = confidence interval, SD = standard deviation.

**Table 4 pathogens-14-00376-t004:** Logistic regression analysis for factors associated with mortality.

	Patients, No. (%)			
Clinical and Laboratory Factors	Alive (n = 137)	Dead (n = 10)	*p*	Odds Ratio (95% CI)
Age ≥ 70 years	55 (40)	8 (80)	0.027	5.96 (1.22–29.15)
ICU admission	32 (23)	9 (90)	0.002	29.53 (3.60–242.02
Severe case	54 (39)	9 (90)	0.014	13.83 (1.70–112.31)
Hospital stay > 14 days	8 (6)	3 (30)	0.013	6.91 (1.50–31.90)
Acute change in mental status	22 (16)	7 (70)	<0.001	12.20 (2.93–50.83)
Acute kidney injury	42 (31)	9 (90)	0.005	20.36 (2.50–165.85)
Need for hemodialysis	6 (4)	5 (50)	<0.001	21.83 (4.94–96.42)
Chest X-ray abnormalities	46 (34)	9 (90)	0.008	17.22 (2.12–140.11)
Lung infiltrates	25 (18)	7 (70)	0.004	24.08 (2.83–205.10)
Lung infiltrates and pleural effusion	4 (3)	2 (20)	0.005	43.00 (3.19–579.75)
Severe dyspnea	5 (4)	6 (60)	<0.001	39.60 (8.42–186.30)
Received antibiotics for pneumonia	21 (15)	5 (50)	0.012	5.48 (1.46–20.58)
Intubation	4 (3)	7 (70)	<0.001	77.58 (14.48–415.80)
Signs of acute heart failure	6 (4)	5 (50)	<0.001	26.67 (5.67–125.49)
	**Patients, Means ± SD**			
**Clinical and Laboratory factors**	**Alive (n = 137)**	**Dead (n = 10)**	** *p* **	**Odds Ratio (95% CI)**
Age, years	63.1 ± 16.1	77.1 ± 10.8	0.01	1.08 (1.02–1.15)
Lactate dehydrogenase (U/L)	909.4 ± 549.7	1767.7 ± 831.0	<0.001	1.002 (1.001–1.003)
Total bilirubin (mg/dL)	2.7 ± 3.2	9.1 ± 11.5	0.003	1.15 (1.05–1.26)
Direct bilirubin (mg/dL)	1.4 ± 2.6	6.7 ± 9.2	0.007	1.19 (1.05–1.36)
Alanine transaminase (U/L)	80.4 ± 69.7	134.0 ± 97.4	0.05	1.006 (1.000–1.013)
Aspartate aminotransferase (U/L)	93.2 ± 75.9	217.5 ± 154.7	<0.001	1.009 (1.004–1.015)
Neutrophils (%)	62.3 ± 15.9	74.3 ± 10.7	0.038	1.074 (1.004–1.149)
Creatinine (mg/dL)	1.2 ± 0.8	3.3 ± 2.6	<0.001	2.37 (1.47–3.82)

No. = number of patients, CI = confidence interval, SD = standard deviation.

## Data Availability

De-identified data supporting the reported results of the study can be requested as needed from the corresponding author.

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
