# Peer review of "Pulmonary Manifestations of Babesiosis and Predictors of Mortality from a Quaternary Care Center in Westchester, New York"

_pathogens, 2025, doi:10.3390/pathogens14040376_

Round 1
Reviewer 1 Report
Comments and Suggestions for Authors
The manuscript describes the clinical manifestations of human babesiosis in patients hospitalized at Westchester Medical Center (New York, USA) between 2006 and 2023; the authors aimed to find predictors of severe diseases and mortality. The clinical manifestations were described in detail; however, the Introduction and Discussion sections require revision.
- Neither the title nor the abstract indicate the state or even the country where the study was conducted.
- Human babesiosis is widespread in the United States, but the authors only cite 18 references, 6 of which were dated before 2000.
- Lines 36-37. References 2-4 are not entirely relevant.
- The introduction lacks epidemiological data on human babesiosis in the United States.
- Line 80 – explain ICU at first mention.
- Line 93. Please provide criteria for distinguishing between the group defined as severely disease and the group defined as non-severely disease.
- Table 1. Because SI units (International System of Units) should be used, temperatures should be reported in degrees Celsius rather than Fahrenheit.
- Line 103. Please provide a map, including the location of counties in New York State.
- Lines 190-191 “In the elderly and patients with comorbidities the infection may result in significant morbidities and even death in up to 6.5 % [7,12].”- the cited articles are from 1992 and 1998. Please refer to more recent articles.
- Lines 215-217 “The rate of this complication (acute kidney injury) had been variable in the literature with a range of 4.3 to 20.4% of reported renal failure …[7,13].- more articles should be cited if they discuss such different frequency of this complication.
- Lines 256-258 “In our study, immunosuppression and splenectomy were not found to be associated with severe disease or increased mortality which was similar to the findings on the mutivariate analysis reported by White et al [7].” – this association is generally recognized. It was noted at least in the cited articles [6] and [17].
- Overall, the Discussion section raises the most questions. For greater clarity of understanding, the first paragraph (1.5 pages long) should be broken into several paragraphs. The authors compared their results with data from other studies; however, throughout the Discussion section, each statement was compared with the results of only 1-2 publications. The authors should review more articles to more fully argue their findings.
- I was surprised by the low parasitemia rate (12% in severe patients). Please discuss whether such a rate has been observed in other studies.
- It was written that spleen rupture is associated with severe infection. Please explain this point. Can severe babesiosis cause spleen rupture?
- The study covers patients from 2006 to 2023. It would be interesting to find out whether the incidence of severe cases and mortality was similar over this long period of time or changed significantly? Was there any evidence that post-COVID syndrome could influence the severity of infection, primarily related to pulmonary symptoms?
Reviewer 2 Report
Comments and Suggestions for Authors
This retrospective study delivers a valuable perspective into the pulmonary manifestations of babesiosis and predictors of mortality in hospitalized patients. The cohort is robust, and the study is clinically relevant, primarily emphasizing respiratory involvement as a severe disease marker. However, the definition of “pulmonary manifestations” could be more clearly outlined, and overlapping criteria for defining severe disease and outcome variables present some bias. Consider adding a more detailed definition of “pulmonary manifestations” at the beginning. The discussion is slightly repetitive of the results, my advice is to focus more on practical advice for clinicians on when to suspect and test for babesiosis. Despite these limitations, the findings provide insight into the manifestations of severe babesiosis and have practical implications for clinicians in endemic areas.
